# Original Solution of Coupled Nonlinear Schrödinger Equations for Simulation of Ultrashort Optical Pulse Propagation in a Birefringent Fiber

**Airat Zhavdatovich Sakhabutdinov** [1,*], **Vladimir Ivanovich Anfinogentov** [1],
**Oleg Gennadievich Morozov** [1], **Vladimir Alexandrovich Burdin** [2],
**Anton Vladimirovich Bourdine** [2,3], **Ildaris Mudarrisovich Gabdulkhakov** [1] and
**Artem Anatolievich Kuznetsov** [1]

[1]  Department of Radiophotonics and Microwave Theory, Kazan National Research State University named
    after A.N. Tupolev-KAI, 31/7, Karl Marx street, Kazan, Rep. Tatarstan 420111, Russia;
    VIAnfinogentov@kai.ru (V.I.A.); microoil@mail.ru (O.G.M.); Baikerboy@mail.ru (I.M.G.);
    aakuznetsov@kai.ru (A.A.K.)
[2]  Department of Communication Lines, Povozhskiy State University of Telecommunications and Informatics,
    23, Lev Tolstoy street, Samara 443010, Russia; burdin@psati.ru (V.A.B.); bourdine@yandex.ru (A.V.B.)
[3]  JSC "Scientific Production Association State Optical Institute Named after Vavilov S.I.", 36/1,
    Babushkin street, Saint Petersburg 192171, Russia
*   Correspondence: azhsakhabutdinov@kai.ru; Tel.: +7-987-290-1864

**Abstract:** This paper discusses approaches to the numerical integration of the coupled nonlinear Schrödinger equations system, different from the generally accepted approach based on the method of splitting according to physical processes. A combined explicit/implicit finite-difference integration scheme based on the implicit Crank–Nicolson finite-difference scheme is proposed and substantiated. It allows the integration of a nonlinear system of equations with a choice of nonlinear terms from the previous integration step. The main advantages of the proposed method are: its absolute stability through the use of an implicit finite-difference integration scheme and an integrated mechanism for refining the numerical solution at each step; integration with automatic step selection; performance gains (or resolutions) up to three or more orders of magnitude due to the fact that there is no need to produce direct and inverse Fourier transforms at each integration step, as is required in the method of splitting according to physical processes. An additional advantage of the proposed method is the ability to calculate the interaction with an arbitrary number of propagation modes in the fiber.

**Keywords:** nonlinear Schrödinger equations system; birefringent fiber; few-mode propagation; Kerr effect; Raman scattering; dispersion; implicit/explicit Crank–Nicolson scheme

## 1. Introduction

Femtosecond lasers hold a strong position in the current industrial production of materials and different-purpose products [1–4]. It should be noted that the problem of the delivery of high-power optic pulses with the required parameters to the destination point appears straight at the beginning of their practical usage. Special optical fibers (e.g., photonic band gap hollow-core and hole-core fibers) are developed for transmission of high-power ultrashort pulses [5–7]. Special attention is paid to polarization maintaining fibers [6]. At the same time, the widespread use of femtosecond lasers requires the usage of cheaper fibers with more simple production techniques. The usage of a shorter pulse technique allows the transmission of pulses with higher peak-power through quartz fibers without maximum available energy exceedance, which can lead to fiber-core degradation [8].

As a result, the cheaper quartz fibers have occupied the niche in industrial usage of femtosecond lasers [9–13]. The birefringent fibers are of particular interest for transmission of high-power ultrashort pulses [10–12], as it has been noted above.

In order to develop and create the methods and delivery technique of ultrashort (femtosecond) pulses it is required to develop mathematical methods of ultrashort pulses evolution modelling in fiber. The ultrashort pulse evolution during its propagation in fiber is described by the coupled nonlinear Schrödinger equations system. The difference between the system mentioned above and the classic form of Schrödinger equations is in the additional terms, which describe the third-order chromatic dispersion and the Raman scattering [13–20]. Taha T.R. and Ablowitz M.I. made the comparative analysis of the currently known numerical methods for solving the nonlinear Schrödinger equation in 1984 [12]. In their fundamental review, they examined many different algorithms, including numerical ones, for solving the nonlinear Schrödinger equation. After this publication, the usage of splitting the physical processes method with the fast Fourier transform became the main numerical method for solving the optics problems in fiber. In particular, it was mentioned that the splitting into physical processes method significantly exceeds the finite difference methods in accuracy, since the second time derivative in it is calculated by the discrete Fourier transform, which provides an exponential convergence rate with respect to the time variable [12]. The split-step Fourier method is used as a standard method in most computer program packages. Although this method has sufficient accuracy, it has its own computing complexity on its non-linear step. It is a good reason to search alternative solution methods, including numerical ones, that can be faster than a split-step Fourier method in case of a large number of time divisions [17,18,21,22].

The experimental results of 12 fs and 175 kW peak-power pulse transmission through birefringent single mode fiber are presented in the series of works [23–28]. The comparison of the experimental results with the computer calculations on fiber end output, which are received using finite difference time domain method, are also presented there. The experimental results correspond to computer calculations in part of pulse duration and its spectrum width. However, the finite difference time domain method does not consider birefringent effects in single mode fiber. At the same time, the computing pulse form at the fiber end output differs from the pulse experimental form significantly. Later in [29–31] it was shown, that the main reason of such discrepancy was connected with the fact that the birefringent effects were not taken into consideration.

Thus, the findings that it is necessary to consider the birefringent effects in fibers, even if fiber length is small [32–34], were confirmed. In the case of the fiber's birefringent effects the system of two Schrödinger equations describes a pulse evolution through fiber. This equations system for fibers with birefringent effects and without Raman scattering is named as the Manakov equations system [20,32].

The system of nonlinear Schrödinger equations has been intensively studied over recent years. Hardin R. and Tappert F. in 1973 [35], as well as Lake B. and co-authors in 1977 [36] were the first, who applied methods of numerical solutions to nonlinear Schrödinger equation solution. Currently, there are many numerical methods for solving the system of coupled nonlinear Schrödinger equations: finite-difference schemes [37,38], spectral methods [39], Petrov-Galerkin method [40], and splitting methods [41–44].

It is known that, it is necessary to take into consideration the third-order chromatic dispersion and Raman scattering for pulses, shorter than 10 ps. As it has been shown above, this leads to the necessity to include the additional terms in the Schrödinger equations. In contrast to [32] it is necessary to solve the modification of coupled a nonlinear Schrödinger equations system [31,33,45]. The principal difference and the major problem here is that nonlinear additional terms contain partial derivatives from desired function by time. The solution with split-step method requires the increase in number of operations in the fast Fourier step or the solving of an additional system of differential equations at each integration step [45]. The projection operator method, based on a variational approach, is suggested in [33] in contrast to split-step methods [31,45].

In this work the numerical integration method is proposed for solving of the coupled nonlinear Schrödinger equations system, written with third-order chromatic dispersion and Raman scattering. The suggested method differs from the generally accepted approach, based on the method of splitting according to physical variables. The system of equations is written in finite-difference relations with separation to linear and nonlinear terms. Linear terms are written in an implicit scheme, and nonlinear terms in an explicit finite-difference scheme. This approach allows researchers to divide the system of Schrödinger equations into two independent systems of linear equations for each mode at each step of numerical integration process. The algorithm for refining the numerical solution at each step is proposed. It eliminates the errors associated with nonlinear terms in explicit form.

The main advantages of the proposed method are the following: absolute stability due to the usage of an implicit finite-difference integration scheme and an integrated mechanism for refining the numerical solution at each step; integration with automatic step selection; increase in the efficiency (or resolutions) up to three or more orders of magnitude due to the fact that there is no need to produce direct and inverse Fourier transforms at each integration step, as is required in a split-step method. An additional advantage of the proposed method is the ability to calculate the interaction with an arbitrary number of propagation modes in the fiber.

## 2. Coupled Nonlinear Schrödinger Equations System

In general terms, the evolution of short optical impulses in a birefringent fiber can be described by the coupled nonlinear Schrödinger equations system:

$$
\begin{cases}
\frac{\partial A_i}{\partial z} = -\frac{\alpha_i}{2} A_i - \beta_{1,i} \frac{\partial A_i}{\partial t} - j \frac{\beta_{2,i}}{2} \frac{\partial^2 A_i}{\partial t^2} + \frac{\beta_{3,i}}{6} \frac{\partial^3 A_i}{\partial t^3} + \\
+ j\gamma_i A_i \sum\limits_{m=1}^{M} C_{i,m} |A_m|^2 - \frac{\gamma_i}{\omega_{0,i}} \sum\limits_{m=1}^{M} B_{i,m} \frac{\partial}{\partial t}\left(|A_m|^2 A_i\right) - j\gamma_i T_R A_i \sum\limits_{m=1}^{M} B_{i,m} \frac{\partial}{\partial t}\left(|A_m|^2\right), \ i = \overline{1,N},
\end{cases}
\quad (1)
$$

where $A_i$—complex envelopes of the optical impulse of the $i$-th mode, $\alpha_i$—attenuation coefficient of the $i$-th mode; $\beta_{1,i}$, $\beta_{2,i}$, $\beta_{3,i}$—the first, second and third order dispersion parameters of the $i$-th mode respectively; $\gamma_i$—nonlinearity parameter for the $i$-th mode; $C_{i,m}$, $B_{i,m}$—coupling coefficients between the $i$-th and $m$-th modes; $T_R$—Raman scattering parameter; $\omega_{0,i}$—angular frequency of the $i$-th mode; $z$—coordinate along the axis of the fiber; $t$—time.

Coupled nonlinear Schrödinger Equations System for two orthogonally polarized modes ($A_X$ and $A_Y$), propagating in a birefringent fiber, which is used for modeling of short optical pulses propagation, has a form equivalent to Equation (1):

$$
\begin{cases}
\frac{\partial A_X}{\partial z} = -\frac{\alpha}{2} A_X - \beta_{1,x} \frac{\partial A_X}{\partial t} - j \frac{\beta_{2,x}}{2} \frac{\partial^2 A_X}{\partial t^2} + \frac{\beta_{3,x}}{6} \frac{\partial^3 A_X}{\partial t^3} + \\
+ j\gamma_x A_X\left(|A_X|^2 + \frac{2}{3}|A_Y|^2\right) - \frac{\gamma_x}{\omega_0} \frac{\partial}{\partial t}\left[|A_X|^2 A_X + \frac{1}{3}|A_Y|^2 A_X\right] - j\gamma_x T_R A_X \frac{\partial}{\partial t}\left(|A_X|^2 + \frac{1}{3}|A_Y|^2\right) \\
\frac{\partial A_Y}{\partial z} = -\frac{\alpha}{2} A_Y - \beta_{1,y} \frac{\partial A_Y}{\partial t} - j \frac{\beta_{2,y}}{2} \frac{\partial^2 A_Y}{\partial t^2} + \frac{\beta_{3,y}}{6} \frac{\partial^3 A_Y}{\partial t^3} + \\
+ j\gamma_y A_Y\left(|A_Y|^2 + \frac{2}{3}|A_X|^2\right) - \frac{\gamma_y}{\omega_0} \frac{\partial}{\partial t}\left[|A_Y|^2 A_Y + \frac{1}{3}|A_X|^2 A_Y\right] - j\gamma_y T_R A_Y \frac{\partial}{\partial t}\left(|A_Y|^2 + \frac{1}{3}|A_X|^2\right)
\end{cases}
\quad (2)
$$

## 3. Initial Conditions and Boundary Terms

For the given the simplest initial conditions, that describe the absence of light in the fiber at the initial and final point of time, terms are described as following:

$$
A_X(0,z) = 0, \ \frac{\partial A_X(0,z)}{\partial t} = 0, \ A_X(T,z) = 0, A_Y(0,z) = 0, \ \frac{\partial A_Y(0,z)}{\partial t} = 0, \ A_Y(T,z) = 0, \ \forall z \in [0,L], \quad (3)
$$

where $T$—final time.

Boundary conditions at one of the fiber ends are described as following time dependent functions:

$$
A_X(t,0) = f_x(t), \ A_Y(t,0) = f_y(t), \quad (4)
$$

## 4. Dimensionless Equations

The system of Equation (1) has the dimension $[W^{\frac{1}{2}}m^{-1}]$, hence the dimensions of the equations constants are:

$$[A_X] = [A_Y] = [\sqrt{W}], \ [\alpha] = \frac{1}{[m]}, \ [\beta_{1,x}] = [\beta_{1,y}] = \frac{[s]}{[m]}, \ [\beta_{2,x}] = [\beta_{2,y}] = \frac{[s^2]}{[m]},$$
$$[\beta_{3,x}] = [\beta_{3,y}] = \frac{[s^3]}{[m]}, \ [\gamma_x] = [\gamma_y] = \frac{1}{[W][m]}, \ [\omega_0] = \frac{1}{[s]}, \ [T_R] = [s]. \tag{5}$$

To transfer Equation (2) into dimensionless form, we involve character process values, namely $L_a$—characteristic length, $T_a$—time, and $P_a$—power:

$$\xi = \frac{z}{L_a}, \ \tau = \frac{t}{T_a}, \ x = \frac{A_X}{\sqrt{P_a}}, \ y = \frac{A_Y}{\sqrt{P_a}}, \tag{6}$$

Replacement of variables in Equation (2) transforms it to dimensionless system:

$$\begin{cases} \frac{\partial x}{\partial \xi} = -\frac{\alpha L}{2}x - \beta_{1,x}\frac{L}{T}\frac{\partial x}{\partial \tau} - j\frac{\beta_{2,x}}{2}\frac{L}{T^2}\frac{\partial^2 x}{\partial \tau^2} + \frac{\beta_{3,x}}{6}\frac{L}{T^3}\frac{\partial^3 x}{\partial \tau^3} + \\ \quad + j\gamma_x LPx\left(|x|^2 + \frac{2}{3}|y|^2\right) - \frac{\gamma_x}{\omega_0}\frac{L}{T}P\frac{\partial}{\partial \tau}\left(|x|^2x + \frac{1}{3}|y|^2x\right) - j\gamma_x L\frac{T_R}{T}Px\frac{\partial}{\partial \tau}\left(|x|^2 + \frac{1}{3}|y|^2\right) \\ \frac{\partial y}{\partial \xi} = -\frac{\alpha L}{2}y - \beta_{1,y}\frac{L}{T}\frac{\partial y}{\partial \tau} - j\frac{\beta_{2,y}}{2}\frac{L}{T^2}\frac{\partial^2 y}{\partial \tau^2} + \frac{\beta_{3,y}}{6}\frac{L}{T^3}\frac{\partial^3 y}{\partial \tau^3} + \\ \quad + j\gamma_y LPy\left(|y|^2 + \frac{2}{3}|x|^2\right) - \frac{\gamma_y}{\omega_0}\frac{L}{T}P\frac{\partial}{\partial \tau}\left(|y|^2y + \frac{1}{3}|x|^2y\right) - j\gamma_y L\frac{T_R}{T}Py\frac{\partial}{\partial \tau}\left(|y|^2 + \frac{1}{3}|x|^2\right) \end{cases} \tag{7}$$

Conversion of the dimensional coefficients into dimensionless is performed according to the formulas:

$$a = \frac{\alpha L_a}{2}, \ b_{1,x} = \beta_{1,x}\frac{L_a}{T_a}, \ b_{2,x} = \frac{\beta_{2,x}}{2}\frac{L_a}{T_a^2}, \ b_{3,x} = \frac{\beta_{3,x}}{6}\frac{L_a}{T_a^3}, \ b_{1,y} = \beta_{1,y}\frac{L_a}{T_a}, \ b_{2,y} = \frac{\beta_{2,y}}{2}\frac{L_a}{T_a^2},$$
$$b_{3,y} = \frac{\beta_{3,y}}{6}\frac{L_a}{T_a^3}, \ u_x = \gamma_x L_a P_a, \ v_x = \frac{u_x}{\omega_0 T_a}, \ w_x = u_x\frac{T_R}{T_a}, \ u_y = \gamma_y L_a P_a, \ v_y = \frac{u_y}{\omega_0 T_a}, \ w_y = u_y\frac{T_R}{T_a}. \tag{8}$$

As a result, we obtain the dimensionless coupled nonlinear Schrödinger equations system in a form suitable for numerical integration:

$$\begin{cases} \frac{\partial x}{\partial \xi} = -[a + \Phi 1_x(x,y)]\cdot x - [b_{1,x} + \Phi 2_x(x,y)]\cdot\frac{\partial x}{\partial \tau} - jb_{2,x}\frac{\partial^2 x}{\partial \tau^2} + b_{3,x}\frac{\partial^3 x}{\partial \tau^3} \\ \frac{\partial y}{\partial \xi} = -[a + \Phi 1_y(x,y)]\cdot y - [b_{1,y} + \Phi 2_y(x,y)]\cdot\frac{\partial y}{\partial \tau} - jb_{2,y}\frac{\partial^2 y}{\partial \tau^2} + b_{3,y}\frac{\partial^3 y}{\partial \tau^3} \end{cases}, \tag{9}$$

where the definitions for non-linear components of the equations are:

$$\Phi_{1x}(x,y) = (v_x + jw_x)\cdot\left(\frac{\partial|x|^2}{\partial \tau} + \frac{1}{3}\frac{\partial|y|^2}{\partial \tau}\right) - ju_x\left(|x|^2 + \frac{2}{3}|y|^2\right),$$
$$\Phi_{1y}(x,y) = (v_y + jw_y)\cdot\left(\frac{1}{3}\frac{\partial|x|^2}{\partial \tau} + \frac{\partial|y|^2}{\partial \tau}\right) - ju_y\left(\frac{2}{3}|x|^2 + |y|^2\right),$$
$$\Phi_{2x}(x,y) = v_x\left(|x|^2 + \frac{1}{3}|y|^2\right), \text{ and } \Phi_{2y}(x,y) = v_y\left(\frac{1}{3}|x|^2 + |y|^2\right). \tag{10}$$

Separation of Equation (7) on linear and nonlinear terms allows us to construct a numerical integration algorithm based on finite-difference methods, where all linear terms can be written in implicit form and nonlinear terms can be written in explicit finite-difference form.

## 5. The Finite-Difference Scheme

The modification of the Crank–Nicolson six-point implicit finite-difference scheme [46] up to an eight-point scheme (Figure 1a), allows us to write the third-order partial derivatives with a second order approximation.

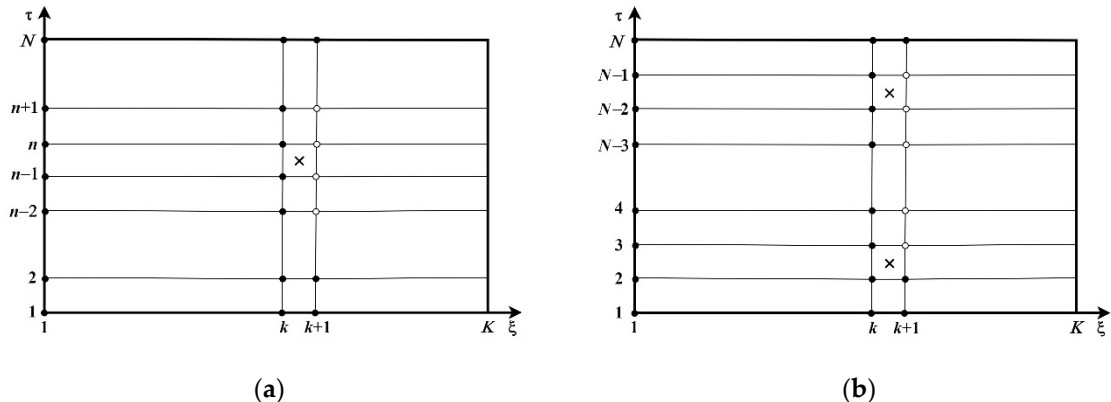

**Figure 1.** Eight-point finite-difference computing scheme for coupled nonlinear Schrödinger equations system integration at: (**a**) internal area; (**b**) boundary.

In Figure 1a, mesh nodes on dimensionless length variable ξ are indicated by *k*, and on dimensionless time τ are indicated by *n*. For Figure 1a, "×" indicates the point, where the equality relations of Equation (9) are recorded. Mesh nodes, where values of desired functions are known or determined, are indicated by "·". If values of desired functions are unknown, they are indicated by "○". Integration is carried out along the length coordinate ξ from left to right. At each integration step, at each point, a system of equations is recorded for four unknown values of the desired functions at the *k* + 1 integration step.

The system of Equation (9) can be rewritten in the form:

$$
\begin{cases}
\frac{\partial x}{\partial \xi} = F_x(x, y) \\
\frac{\partial y}{\partial \xi} = F_y(x, y)
\end{cases},
\tag{11}
$$

where $F_x(x, y)$, $F_y(x, y)$ are used to denote appropriate parts in Equation (9).

If we add the parameter θ, that changes from zero to one, we can rewrite Equation (11) in finite-difference implicit form. The finite-difference equivalences are written in each virtual point with fractional indexes $(k + \frac{1}{2}, n - \frac{1}{2})$. This point is denoted in Figure 1a by "×". The finite-difference equation system has the form:

$$
\begin{cases}
\frac{(x_{k+1}^n + x_{k+1}^{n-1}) - (x_k^n + x_k^{n-1})}{2\Delta\xi} = \theta \cdot (F_x)_{k+1}^{n-1/2} + (1-\theta) \cdot (F_x)_k^{n-1/2} \\
\frac{(y_{k+1}^n + y_{k+1}^{n-1}) - (y_k^n + y_k^{n-1})}{2\Delta\xi} = \theta \cdot (F_y)_{k+1}^{n-1/2} + (1-\theta) \cdot (F_y)_k^{n-1/2}
\end{cases},
\tag{12}
$$

where bottom indexes "*k*" are used to denote the discrete dimensionless length, and top "*n*" indexes—for dimensionless time.

The values of desired functions in Equation (12) are known at the *k*-s layer, while the values at the (*k* + 1) layer are known only on the boundary. The values of $(F_x)_k^{n-1/2}$, $(F_y)_k^{n-1/2}$ are attributed to virtual mesh node $(k, n - \frac{1}{2})$, and $(F_x)_{k+1}^{n-1/2}$, $(F_y)_{k+1}^{n-1/2}$ are attributed to virtual mesh node $(k + 1, n - \frac{1}{2})$. Dependence of θ parameter equivalences in Equation (12) will be written in virtual node with indexes $(k + \theta, n - \frac{1}{2})$.

The nonlinearity, presented in $\Phi_{1x}(x, y)$, $\Phi_{1x}(x, y)$, $\Phi_{2y}(x, y)$, $\Phi_{2y}(x, y)$, and a third-order partial derivative by time, do not allow the use of the classic approach to equation system solutions (Thomas, or tridiagonal matrix, algorithm). The modification of Crank–Nicolson computing scheme is offered in [46]. The main idea is to write all linear terms in implicit form, and all nonlinear terms in explicit

form. Using the recommendations given in [46], we write the expressions for $F_x(x, y)$ and $F_y(x, y)$ as the sum of linear and nonlinear parts:

$$(F_x)_k^{n-1/2} = (L_x)_k^{n-1/2} + (N_x)_k^{n-1/2}, \quad (F_y)_k^{n-1/2} = (L_y)_k^{n-1/2} + (N_y)_k^{n-1/2}, \tag{13}$$

where the definition for linear parts are:

$$\begin{aligned}
(L_x)_k^{n-1/2} &= -a x_k^{n-1/2} - b_{1,x}\left(\frac{\partial x}{\partial \tau}\right)_k^{n-1/2} - jb_{2,x}\left(\frac{\partial^2 x}{\partial \tau^2}\right)_k^{n-1/2} + b_{3,x}\left(\frac{\partial^3 x}{\partial \tau^3}\right)_k^{n-1/2}, \\
(L_y)_k^{n-1/2} &= -a y_k^{n-1/2} - b_{1,y}\left(\frac{\partial y}{\partial \tau}\right)_k^{n-1/2} - jb_{2,y}\left(\frac{\partial^2 y}{\partial \tau^2}\right)_k^{n-1/2} + b_{3,y}\left(\frac{\partial^3 y}{\partial \tau^3}\right)_k^{n-1/2},
\end{aligned} \tag{14}$$

and for nonlinear parts:

$$\begin{aligned}
(N_x)_k^{n-1/2} &= -(\Phi_{1x})_k^{n-1/2} x_k^{n-1/2} - (\Phi_{2x})_k^{n-1/2}\left(\frac{\partial x}{\partial \tau}\right)_k^{n-1/2}, \\
(N_y)_k^{n-1/2} &= -(\Phi_{1y})_k^{n-1/2} y_k^{n-1/2} - (\Phi_{2y})_k^{n-1/2}\left(\frac{\partial y}{\partial \tau}\right)_k^{n-1/2}.
\end{aligned} \tag{15}$$

To write the linear terms on $(k + 1)$-th layer, it is enough to replace «$k$» on «$k + 1$» in Equation (14). For nonlinear terms on $(k + 1)$-th layer, it is necessary to use explicit finite-difference definition, using values from the previous integration layer:

$$\begin{aligned}
(N_x)_{k+1}^{n-1/2} &= -(\Phi_{1x})_k^{n-1/2} x_{k+1}^{n-1/2} - (\Phi_{2x})_k^{n-1/2}\left(\frac{\partial x}{\partial \tau}\right)_{k+1}^{n-1/2}, \\
(N_y)_{k+1}^{n-1/2} &= -(\Phi_{1y})_k^{n-1/2} y_{k+1}^{n-1/2} - (\Phi_{2y})_k^{n-1/2}\left(\frac{\partial y}{\partial \tau}\right)_{k+1}^{n-1/2}.
\end{aligned} \tag{16}$$

Notably, the explicit form in Equation (16) is taken only for $\Phi_{1x}(x, y)$, $\Phi_{2x}(x, y)$, $\Phi_{1y}(x, y)$, $\Phi_{2y}(x, y)$, while for the linear parts of Equation (16), the implicit form will be used.

The values of nonlinear terms in virtual mesh point $(k, n - \frac{1}{2})$ for $\Phi_{1x}(x, y)$ and $\Phi_{1y}(x, y)$ can be written as:

$$\begin{aligned}
(\Phi_{1x})_k^{n-1/2} &= -\frac{ju_x}{2}\left(|x_k^n|^2 + |x_k^{n-1}|^2 + \frac{2}{3}\left(|y_k^n|^2 + |y_k^{n-1}|^2\right)\right) + \frac{v_x + jw_x}{\Delta\tau}\left(|x_k^n|^2 - |x_k^{n-1}|^2 + \frac{1}{3}\left(|y_k^n|^2 - |y_k^{n-1}|^2\right)\right), \\
(\Phi_{1y})_k^{n-1/2} &= -\frac{ju_y}{2}\left(\frac{2}{3}\left(|x_k^n|^2 + |x_k^{n-1}|^2\right) + |y_k^n|^2 + |y_k^{n-1}|^2\right) + \frac{v_y + jw_y}{\Delta\tau}\left(\frac{1}{3}\left(|x_k^n|^2 - |x_k^{n-1}|^2\right) + |y_k^n|^2 - |y_k^{n-1}|^2\right).
\end{aligned} \tag{17}$$

and for $\Phi_{2x}(x, y)$ and $\Phi_{2y}(x, y)$:

$$\begin{aligned}
(\Phi_{2x})_k^{n-1/2} &= \frac{v_x}{2}\left(|x_k^n|^2 + |x_k^{n-1}|^2 + \frac{1}{3}\left(|y_k^n|^2 + |y_k^{n-1}|^2\right)\right), \\
(\Phi_{2y})_k^{n-1/2} &= \frac{v_y}{2}\left(\frac{1}{3}\left(|x_k^n|^2 + |x_k^{n-1}|^2\right) + |y_k^n|^2 + |y_k^{n-1}|^2\right).
\end{aligned} \tag{18}$$

The desired function values in virtual mesh point $(k, n - \frac{1}{2})$ are written as a half of its sum in neighbor mesh points:

$$x_k^{n-1/2} = \frac{x_k^n + x_k^{n-1}}{2}, \ x_{k+1}^{n-1/2} = \frac{x_{k+1}^n + x_{k+1}^{n-1}}{2}, \ y_k^{n-1/2} = \frac{y_k^n + y_k^{n-1}}{2}, \ y_{k+1}^{n-1/2} = \frac{y_{k+1}^n + y_{k+1}^{n-1}}{2}. \tag{19}$$

The partial derivative from desired functions on dimensionless time in the mesh points $(k, n - \frac{1}{2})$ are written as central finite-differences with second-order accuracy:

$$\left(\frac{\partial x}{\partial \tau}\right)_k^{n-1/2} = \frac{x_k^n - x_k^{n-1}}{\Delta\tau}, \ \left(\frac{\partial^2 x}{\partial \tau^2}\right)_k^{n-1/2} = \frac{x_k^{n+1} - x_k^n - x_k^{n-1} + x_k^{n-2}}{2\Delta\tau^2}, \ \left(\frac{\partial^3 x}{\partial \tau^3}\right)_k^{n-1/2} = \frac{x_k^{n+1} - 3x_k^n + 3x_k^{n-1} - x_k^{n-2}}{\Delta\tau^3}. \tag{20}$$

The partial derivatives for desired mesh function $y$ are written similarly. In order to write the $x$ and $y$ in the mesh nodes $(k + 1, n - \frac{1}{2})$, it is enough to replace "$k$" with "$k + 1$" in Equation (20).

　　　We substitute the expressions of Equations (20)–(13) in Equation (12), then we regroup terms (all known variables to the right side and all unknown variables to the left side of equations), and denote right (known) side of equations as:

$$
\begin{aligned}
(R_X)_{k+1}^n &= \frac{x_k^n + x_k^{n-1}}{2 \cdot \Delta \xi \cdot \theta} + \frac{(1-\theta)}{\theta}\left((L_x)_k^{n-1/2} + (N_x)_k^{n-1/2}\right), \\
(R_Y)_{k+1}^n &= \frac{y_k^n + y_k^{n-1}}{2 \cdot \Delta \xi \cdot \theta} + \frac{(1-\theta)}{\theta}\left((L_y)_k^{n-1/2} + (N_y)_k^{n-1/2}\right),
\end{aligned}
\tag{21}
$$

where the linear and nonlinear terms are defined in Equations (14) and (15), then the equation system will be as follows:

$$
\begin{cases}
\frac{x_{k+1}^n + x_{k+1}^{n-1}}{2 \cdot \Delta \xi \cdot \theta} + \frac{a + (\Phi_{1x})_k^{n-1/2}}{2}\left(x_{k+1}^n + x_{k+1}^{n-1}\right) + \frac{b_{1,x} + (\Phi_{2x})_k^{n-1/2}}{\Delta \tau}\left(x_{k+1}^n - x_{k+1}^{n-1}\right) + \\
+ \frac{jb_{2,x}}{2\Delta\tau^2}\left(x_{k+1}^{n+1} - x_{k+1}^n - x_{k+1}^{n-1} + x_{k+1}^{n-2}\right) - \frac{b_{3,x}}{\Delta\tau^3}\left(x_{k+1}^{n+1} - 3x_{k+1}^n + 3x_{k+1}^{n-1} - x_{k+1}^{n-2}\right) = (R_X)_{k+1}^n \\
\frac{y_{k+1}^n + y_{k+1}^{n-1}}{2 \cdot \Delta \xi \cdot \theta} + \frac{a + (\Phi_{1y})_k^{n-1/2}}{2}\left(y_{k+1}^n + y_{k+1}^{n-1}\right) + \frac{b_{1,y} + (\Phi_{2y})_k^{n-1/2}}{\Delta \tau}\left(y_{k+1}^n - y_{k+1}^{n-1}\right) + \\
+ \frac{jb_{2,y}}{2\Delta\tau^2}\left(y_{k+1}^{n+1} - y_{k+1}^n - y_{k+1}^{n-1} + y_{k+1}^{n-2}\right) - \frac{b_{3,y}}{\Delta\tau^3}\left(y_{k+1}^{n+1} - 3y_{k+1}^n + 3y_{k+1}^{n-1} - y_{k+1}^{n-2}\right) = (R_Y)_{k+1}^n
\end{cases}
\tag{22}
$$

　　　We rewrite the Equation (22) in canonical line equation systems form:

$$
\begin{cases}
A_x^n x_{k+1}^{n-2} + B_x^n x_{k+1}^{n-1} + C_x^n x_{k+1}^n + D_x^n x_{k+1}^{n+1} = (R_X)_{k+1}^n \\
A_y^n y_{k+1}^{n-2} + B_y^n y_{k+1}^{n-1} + C_y^n y_{k+1}^n + D_y^n y_{k+1}^{n+1} = (R_Y)_{k+1}^n
\end{cases}
\tag{23}
$$

　　　The *A*, *B*, *C* and *D* coefficients in Equation (23) for desired mesh function *x* are determined by formulas:

$$
\begin{aligned}
A_x^n &= \frac{jb_{2,x}}{2\Delta\tau^2} + \frac{b_{3,x}}{\Delta\tau^3}, \\
B_x^n &= \frac{1}{2 \cdot \Delta \xi \cdot \theta} + \frac{a + (\Phi_{1x})_k^{n-1/2}}{2} - \frac{b_{1,x} + (\Phi_{2x})_k^{n-1/2}}{\Delta\tau} - \frac{jb_{2,x}}{2\Delta\tau^2} - 3\frac{b_{3,x}}{\Delta\tau^3}, \\
C_x^n &= \frac{1}{2 \cdot \Delta \xi \cdot \theta} + \frac{a + (\Phi_{1x})_k^{n-1/2}}{2} + \frac{b_{1,x} + (\Phi_{2x})_k^{n-1/2}}{\Delta\tau} - \frac{jb_{2,x}}{2\Delta\tau^2} + 3\frac{b_{3,x}}{\Delta\tau^3}, \\
D_x^n &= \frac{jb_{2,x}}{2\Delta\tau^2} - \frac{b_{3,x}}{\Delta\tau^3},
\end{aligned}
\tag{24}
$$

and for mesh function *y*:

$$
\begin{aligned}
A_y^n &= \frac{jb_{2,y}}{2\Delta\tau^2} + \frac{b_{3,y}}{\Delta\tau^3}, \\
B_y^n &= \frac{1}{2 \cdot \Delta \xi \cdot \theta} + \frac{a + (\Phi_{1y})_k^{n-1/2}}{2} - \frac{b_{1,y} + (\Phi_{2y})_k^{n-1/2}}{\Delta\tau} - \frac{jb_{2,y}}{2\Delta\tau^2} - 3\frac{b_{3,y}}{\Delta\tau^3}, \\
C_y^n &= \frac{1}{2 \cdot \Delta \xi \cdot \theta} + \frac{a + (\Phi_{1y})_k^{n-1/2}}{2} + \frac{b_{1,y} + (\Phi_{2y})_k^{n-1/2}}{\Delta\tau} - \frac{jb_{2,y}}{2\Delta\tau^2} + 3\frac{b_{3,y}}{\Delta\tau^3}, \\
D_y^n &= \frac{jb_{2,y}}{2\Delta\tau^2} - \frac{b_{3,y}}{\Delta\tau^3}.
\end{aligned}
\tag{25}
$$

## 6. Boundary Conditions

　　　The integration process is going from layer to layer according to dimensionless length $\xi$. The system of Equation (23) is being solved on each layer along dimensionless time $\tau$. Boundary conditions of Equation (4) in a finite-difference computing scheme are transformed into initial conditions, so mesh function values at initial ($k = 1$) and at previous ($k$-th) integration layer will be known. Initial conditions of Equations (3) and (4) provide the boundary conditions for function values at $n = 1, 2$, and $N$. The Equations (3) and (4) in dimensionless variables and finite-difference form will be written as:

$$
\begin{aligned}
(x_0)_k^0 &= \frac{X(0, L_a \xi_k)}{\sqrt{P_a}}, & (y_0)_k^0 &= \frac{Y(0, L_a \xi_k)}{\sqrt{P_a}}, \\
(x_0)_k^1 &= (x_0)_k^0 + \frac{T_a \Delta \tau}{\sqrt{P_a}} X'(0, L_a \xi_k), & (y_0)_k^1 &= (y_0)_k^0 + \frac{T_a \Delta \tau}{\sqrt{P_a}} Y'(0, L_a \xi_k), \\
(x_0)_k^N &= \frac{1}{\sqrt{P_a}} X(T_a, L_a \xi_k), & (y_0)_k^N &= \frac{1}{\sqrt{P_a}} Y(T_a, L_a \xi_k), \\
x_0^n &= \frac{1}{\sqrt{P_a}} f_x(T_a \tau_n), & y_0^n &= \frac{1}{\sqrt{P_a}} f_y(T_a \tau_n).
\end{aligned}
\tag{26}
$$

The situational views on the boundary integration area for the eight-point computing scheme are presented in Figure 1b. The nodes of mesh, where values are known or given, are marked by black dots.

It is worth noting that $n$ index in Equation (23) takes values from 3 up to $(N-1)$. The common form of equations has a different form at $n = 3, 4$ и$(N-1)$, because Equation (23) includes the boundary conditions in explicit form. We write them separately:

$$n = 3, \quad \begin{cases} A_x^3 x_{k+1}^1 + B_x^3 x_{k+1}^2 + C_x^3 x_{k+1}^3 + D_x^3 x_{k+1}^4 = (R_X)_{k+1}^3 \\ A_y^3 y_{k+1}^1 + B_y^3 y_{k+1}^2 + C_y^3 y_{k+1}^3 + D_y^3 y_{k+1}^4 = (R_Y)_{k+1}^3 \end{cases} . \tag{27}$$

The values of $x_{k+1}^1$, $x_{k+1}^2$ and $y_{k+1}^1$, $y_{k+1}^2$ in Equation (27) are known, hence, we can transfer these terms into the right part of equations and for $n = 3$ system, Equation (27) can be written as:

$$n = 3, \quad \begin{cases} C_x^3 x_{k+1}^3 + D_x^3 x_{k+1}^4 = (R_X)_{k+1}^3 - A_x^3 x_{k+1}^1 - B_x^3 x_{k+1}^2 \\ C_y^3 y_{k+1}^3 + D_y^3 y_{k+1}^4 = (R_Y)_{k+1}^3 - A_y^3 y_{k+1}^1 - B_y^3 y_{k+1}^2 \end{cases} . \tag{28}$$

and the same for $n = 4$:

$$n = 4, \quad \begin{cases} B_x^4 x_{k+1}^3 + C_x^4 x_{k+1}^4 + D_x^4 x_{k+1}^5 = (R_X)_{k+1}^4 - A_x^4 x_{k+1}^2 \\ B_y^4 y_{k+1}^3 + C_y^4 y_{k+1}^4 + D_y^4 y_{k+1}^5 = (R_Y)_{k+1}^4 - A_y^4 y_{k+1}^2 \end{cases} . \tag{29}$$

If $n = (N-1)$, the $x_{k+1}^N$ and $y_{k+1}^N$ are known, hence:

$$n = N-1, \quad \begin{cases} A_x^{N-1} x_{k+1}^{N-3} + B_x^{N-1} x_{k+1}^{N-2} + C_x^{N-1} x_{k+1}^{N-1} = (R_X)_{k+1}^{N-1} - D_x^{N-1} x_{k+1}^N \\ A_y^{N-1} y_{k+1}^{N-3} + B_y^{N-1} y_{k+1}^{N-2} + C_y^{N-1} y_{k+1}^{N-1} = (R_Y)_{k+1}^{N-1} - D_y^{N-1} y_{k+1}^N \end{cases} . \tag{30}$$

## 7. The Line Equation System in Classic Form

It should be especially noted that Equation (23) breaks down in two independent line equation systems, relative to mesh node variables $x_{k+1}^n$ and $y_{k+1}^n (n = \overline{3, N-1})$. These two equation systems can be solved separately at each integration step along the length.

We can write each line equation system in Equation (23) in matrix from:

$$\mathbf{M} \times \mathbf{X} = \mathbf{R}. \tag{31}$$

The matrix $\mathbf{M}$ is four-diagonal matrix, where in addition to main diagonal, which consists of "C" coefficients, contains one "upper" ("D") and two "sub" ("A" and "B") diagonals. There is no need to store the whole matrix $\mathbf{M}$ in computer memory. Instead, we use the two sets of five arrays for all four diagonals and its right member vector for each part of Equation (23). The $\mathbf{A}$ and $\mathbf{B}$ arrays are used for two sub diagonals storage, $\mathbf{C}$—for main diagonal, $\mathbf{D}$—for upper diagonal and $\mathbf{R}$—for right member vector:

$$\begin{aligned} \mathbf{A}[i] &= M_{i,i-2} &= A_*^i, & 4 \le i \le N-1 \\ \mathbf{B}[i] &= M_{i,i-1} &= B_*^i, & 3 \le i \le N-1 \\ \mathbf{C}[i] &= M_{i,i} &= C_*^i, & 2 \le i \le N-1 \\ \mathbf{D}[i] &= M_{i,i+1} &= D_*^i, & 2 \le i \le N-2 \\ \mathbf{R}[i] &= R_i &= (R_*)_{k+1}^i & 2 \le i \le N-1 \end{aligned} . \tag{32}$$

A syntax form, familiar to many programming languages, is used in Equation (32). It allows us to discern variables $A$, $B$, $C$, and $D$, used in Equation (23) and above, from arrays $\mathbf{A}$, $\mathbf{B}$, $\mathbf{C}$, and $\mathbf{D}$, used for matrix $\mathbf{M}$ elements notation.

The line equation system solution with $m$-diagonal matrix comes down to sequential transforming matrix to an upper triangular matrix and vector of unknowns is computed backwards. We can use our

modification of Thomas tridiagonal algorithm, which allows the transformation of the four-diagonal matrix **M** to the triangular form, according to:

$$
\begin{aligned}
\mathbf{R}[i] &= \mathbf{R}[i+1] - \mathbf{R}[i] \times \mathbf{C}[i+1]/\mathbf{D}[i], \\
\mathbf{A}[i] &= -\mathbf{A}[i] \times \mathbf{C}[i+1]/\mathbf{D}[i], \\
\mathbf{B}[i] &= \mathbf{A}[i+1] - \mathbf{B}[i] \times \mathbf{C}[i+1]/\mathbf{D}[i], \qquad \forall i = \overline{N-2,3,-1}. \\
\mathbf{C}[i] &= \mathbf{B}[i+1] - \mathbf{C}[i] \times \mathbf{C}[i+1]/\mathbf{D}[i], \\
\mathbf{D}[i] &= \mathbf{C}[i+1] - \mathbf{D}[i] \times \mathbf{C}[i+1]/\mathbf{D}[i] = 0,
\end{aligned}
\tag{33}
$$

After the matrix **M** (with right member vector **R**) is transformed, we calculate unknown vectors according to:

$$
\begin{aligned}
\mathbf{T}[3] &= \mathbf{R}[3]/\mathbf{C}[3], \\
\mathbf{T}[4] &= (\mathbf{R}[4] - \mathbf{B}[4] \times \mathbf{T}[3])/\mathbf{C}[4], \\
\mathbf{T}[i] &= (\mathbf{R}[i] - \mathbf{A}[i] \times \mathbf{T}[i-2] - \mathbf{B}[i] \times \mathbf{T}[i-1])/\mathbf{C}[i], \ \forall i = \overline{5, N-1},
\end{aligned}
\tag{34}
$$

where the unknown vector, as in the Equation (23) solution, is denoted as **T**.

## 8. Numerical Solution Refining Algorithm

The refining solution algorithm is used at each integration step. The explicit nonlinear terms determination (from previous $k$-th integration layer) makes its contribution to the inaccuracy of new $(k + 1)$-th layer values calculation. The main idea of the refining algorithm is in the iterative process organization at each integration step, which corrects nonlinear terms. At the first iteration step the numerical solution for mesh functions $x_{k+1}^n$ and $y_{k+1}^n$ is searched according to the method suggested above. At the next integration step we can correct the values of nonlinear terms since we can use values calculated previously from average between $k$-th and $(k + 1)$-th layers to renew the nonlinear terms. The iteration process stops when the absolute difference between two values of desired mesh functions, calculated on two iteration steps, is less than the given allowable error.

## 9. Method Verification on Some Classic Tasks

The coupled nonlinear Schrödinger equations system, written with third-order dispersion and Raman scattering, coincides with various classic equations of mathematical physics. So, we can obtain the classic heat diffusion in a solid rod task with or without convection. In other cases, the coupled nonlinear Schrödinger equations system can coincide with Korteweg–de Vries equation of waves on shallow water surfaces. The coupled Manakov equations system in multimode fibers with strongly (and weakly) coupled groups of modes is also a particular case of the coupled nonlinear Schrödinger equations system [29–31,46].

Each of these equations (except Manakov system) has real desired functions and real variables—time and length. In our case we have two complex desired functions of two real variables—length and time. It is required to swap time and length variables in the coupled nonlinear Schrödinger equations system in order to use it for heat diffusion or the Korteweg–de Vries task applies. Besides, it is required to set all imaginary parts of desired functions equal to zero. For the coupled nonlinear Schrödinger equations system, used as Manakov system, the special set of equation constants is required.

### 9.1. Heat Diffusion in a Solid Rod Task

The coupled nonlinear Schrödinger equations system numerical solution, used as equation of heat diffusion task without heat conductivity [47], is shown in Figure 2a. The equations system variable time $t$ is used as length and equations system variable $z$ is used as physical time (time and length in the equations system are swapped), and the equations system constants are: $\alpha \neq 0$, $\beta_{1,x} = \beta_{1,y} = 0$, $\beta_{2,x} \neq 0$, $\beta_{2,y} = 0$, $\beta_{3,y} = \beta_{3,y} = 0$, $\gamma_x = \gamma_y = 0$.

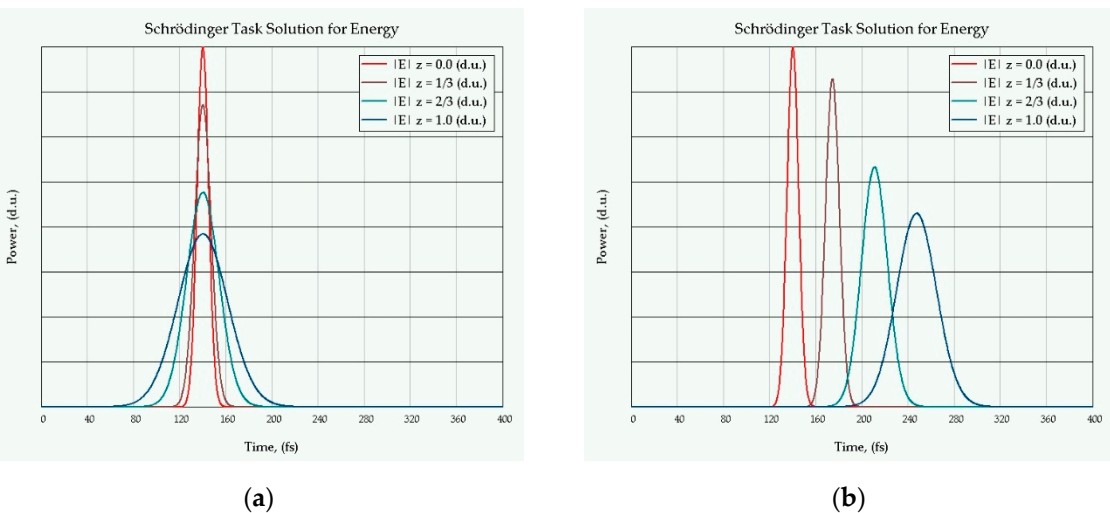

**Figure 2.** Heat diffusion in the solid rod task numerical results: (**a**) without convection; (**b**) with connectivity. Curves for different length values are marked: red at z = 0 (initial), brown at z = 1/3, green at z = 2.3, blue at z = 1.

In Figure 2a we can see that in initial moment of time ($z = 0$) the point with length coordinate ($t = 140$) is heated up. In the process of time (with $z$ growing) it becomes colder in this point, due to $\alpha \neq 0$, and heat diffuses to the left and to the right sides, due to $\beta_{2,x} \neq 0$. If we also request the $\beta_{1,x} \neq 0$, the heat convection appears.

The numerical results at values $\alpha \neq 0$, $\beta_{1,x} \neq 0$, $\beta_{1,y} = 0$, $\beta_{2,x} \neq 0$, $\beta_{2,y} = 0$, $\beta_{3,y} = 0$, $\beta_{3,y} = 0$, $\gamma_x = 0$, $\gamma_y = 0$ are shown in Figure 2b. We can see, that in process of time (with $z$ growing), in addition to previous effects, the heat point is moving along the length (along the $t$ variable). The heat diffusion task solution, based on coupled nonlinear Schrödinger equations system, demonstrates an excellent matching with classic heat diffusion equation solutions, including analytical solutions, by its character and values.

### 9.2. The Korteweg–De Vries and Linear Tasks

The Korteweg–de Vries equation is a mathematical model of waves on shallow water surfaces. It is particularly notable as the prototypical example of an exactly solvable model with non-linear partial differential equation whose solutions can be exactly and precisely specified. If we set the equations system coefficients equal to $\alpha = 0$, $\beta_{1,x} = 0$, $\beta_{1,y} = 0$, $\beta_{2,x} = 0$, $\beta_{2,y} = 0$, $\beta_{3,y} \neq 0$, $\beta_{3,y} = 0$, $\gamma_y = 0$ and choose special and different values $\gamma_x$ for individual nonlinear terms, we receive the Korteweg–de Vries equation. The obtained computing results are presented in Figure 3a. Its common character almost coincides with our previous results [29–31,46], and other authors' results [48].

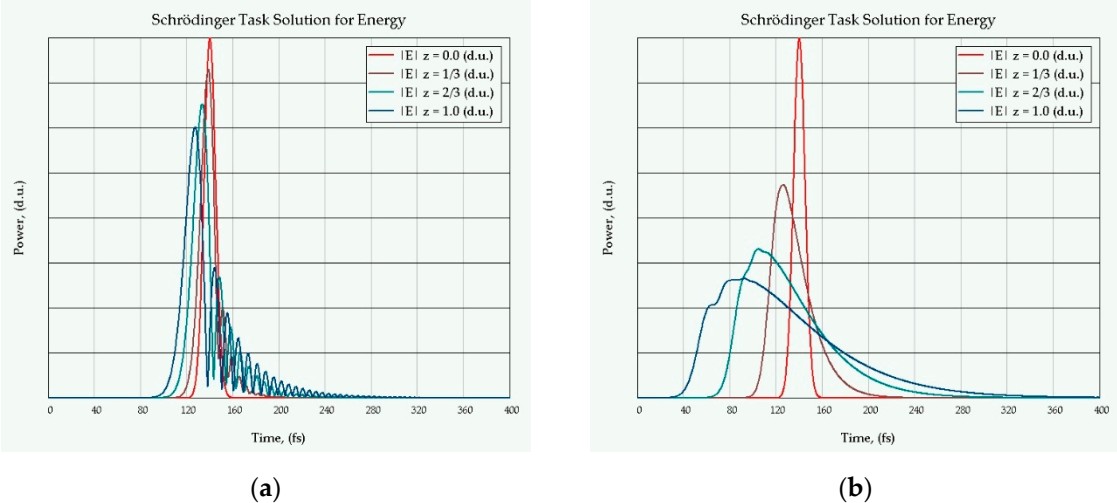

**Figure 3.** The numerical results of: (**a**) Korteweg–de Vries equation; (**b**) coupled linear Schrödinger equations system. Curves for different length values are marked: red at z = 0 (initial), brown at z = 1/3, green at z = 2.3, blue at z = 1.

The computing results of the coupled Schrödinger equations system solution in its linear case (at $\gamma_x = \gamma_y = 0$) are shown in Figure 3b. The obtained numerical results coincide with physical processes as well as with other solutions in the literature [29–31].

*9.3. The Ultra-Short Pulse Evolution in Fiber*

The special case of the coupled nonlinear Schrödinger equations system is the case when this equations system coincides with the Manakov equations system with second-order dispersion and Raman scattering, when $\beta_{3,y} = \beta_{3,y} = 0$, $T_R \to 0$ and $\omega_0 \to \infty$. The computing results of the Manakov equations system task, received on the base of coupled nonlinear Schrödinger equations system, is shown in Figure 4a. These results coincide with our previous results [29–31,46] and other researchers' results [23–28,49].

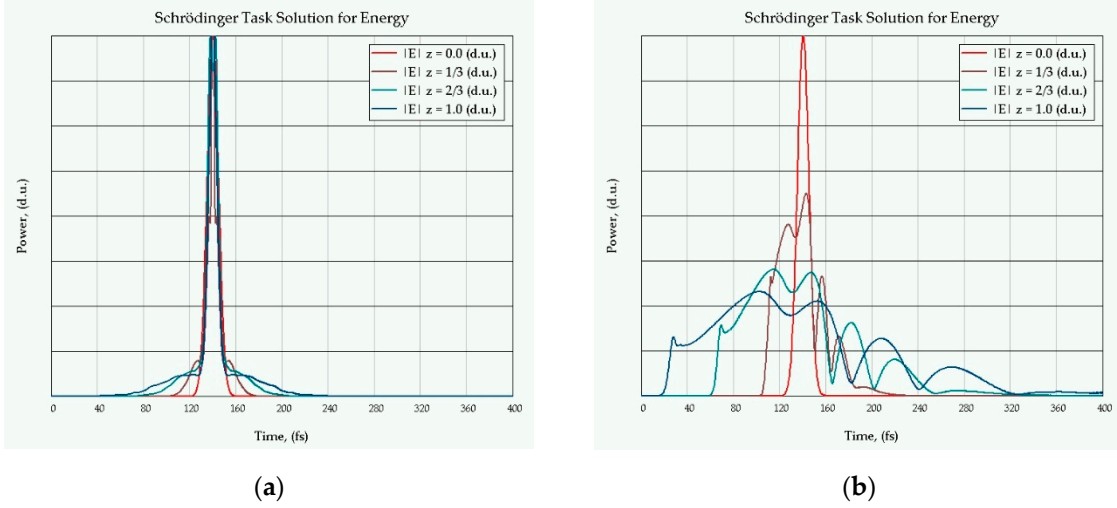

**Figure 4.** The pulse form evolution in fiber of: (**a**) Manakov equations system; (**b**) coupled nonlinear Schrödinger equations system. Curves for different length values are marked: red at z = 0 (initial), brown at z = 1/3, green at z = 2.3, blue at z = 1.

The ultra-short pulse evolution in fiber with third-order dispersion and Raman scattering is described by complete coupled nonlinear Schrödinger equations system. The values from [23–31], which were used in their experiment, were taken as: $\alpha$ = 0.2 dB·m/km, $\beta_{1,x}$ = 4.294 × 10$^{-9}$ s/m, $\beta_{1,y}$ = 4.290 × 10$^{-9}$ s/m, $\beta_{2,x}$ = 3.600 × 10$^{-26}$ s$^2$/m, $\beta_{2,y}$ = 3.250 × 10$^{-26}$ s$^2$/m, $\beta_{3,y}$ = $\beta_{3,y}$ = 2.750 × 10$^{-41}$ s$^3$/m, $\gamma_x$ = $\gamma_y$ = 3.600 × 10$^{-2}$ (m·W)$^{-1}$, $T_R$ = 4.000 × 10$^{-15}$ s, $\omega_0$ = 2.3612 × 10$^{-15}$ s$^{-1}$ (wavelength 798 nm). The single chirped Gauss pulse is in the input fiber end (chirp C = –0.4579), pulse duration is 12 fs, with maximum power $P$ = 1.75 × 10$^5$ W. The pulse form is described as:

$$f(t) \; = \; A \cdot \exp\left(-\frac{(1 - j \cdot C)(t - T)^2}{2 \cdot \tau^2}\right). \tag{35}$$

The pulse evolution according to computing results is shown in Figure 4b. The number of mesh points along dimensionless time was chosen as 20,000; approximately 720,000 integration steps were made along dimensionless time with initial time step $\Delta\xi$ = 1·10$^{-4}$ d.u. Besides, the automatic integration step correction algorithm was included. It allowed the calculation of pulse evolution length up to ~2.5 mm. The maximum error for iteration process was chosen as 10$^{-30}$ d.u. All calculations were made in a processor with double precision and 64-bit architecture.

The computing result curve (blue line in Figure 4b) is in good matching with experimental results, presented in [23–31]. It excellently confirms that the suggested method is effective and can be used to solve similar nonlinear tasks.

## 10. Conclusions

In our research we showed that the suggested method can be successfully used for solving the coupled nonlinear Schrödinger equations system in case of strongly coupled groups of modes for pulse evolution. In comparison with the splitting by physical processes method, the proposed method is absolutely stable due to the usage of an implicit finite-difference integration scheme. Commonly, our method has the following advantages: firstly, computational complexity is reduced, since two (direct and inverse) Fourier transforms are replaced with the numerical linear equations system solution at each integration step; secondly, possible errors in computing schemes are reduced because the nonlinear terms are not taken from the previous layer at each integration step; thirdly, the separation of the nonlinear system of Schrödinger equations into two independent linear equation systems at each integration step allows us to include an arbitrary number of propagation modes into equations and investigate their mutual influence.

Results, received from model task investigations and their comparison with other researcher's results, as well as experimental data, allows us to conclude that the suggested method is effective, advantageous and has potential for future improvement.

**Author Contributions:** Conceptualization, A.Z.S.; Formal analysis, V.I.A., and V.A.B.; Funding acquisition, O.G.M., A.V.B. and A.A.K.; Investigation, A.Z.S., V.I.A., V.A.B., A.A.K., and I.M.G.; Methodology, A.Z.S., V.I.A., O.G.M., and V.A.B.; Software, A.Z.S. and V.I.A.; Supervision, V.A.B., and A.V.B.; Validation, A.Z.S., V.I.A., O.G.M. and V.A.B.; Visualization, A.Z.S.; Writing—review & editing, A.Z.S. All authors have read and agreed to the published version of the manuscript.

**Funding:** I.M.G. was funded by Russian Foundation for Basic Research: 19-37-90057, A.A.K. was funded by President of the Russian Federation for state support of young Russian scientists—347 candidates of sciences MK-3421.2019.8: 075-15-2019-309, A.V.B. and O.G.M was funded by Russian Foundation for Basic Research: 19-57-80006 346 BRICS_t.

**Conflicts of Interest:** The authors declare no conflict of interest.

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
