# Peer review of "Original Solution of Coupled Nonlinear Schrödinger Equations for Simulation of Ultrashort Optical Pulse Propagation in a Birefringent Fiber"

_fibers, doi:10.3390/fib8060034_

Round 1
Reviewer 1 Report
The authors discuss approaches for the numerical integration of the nonlinear Schroedinger equation. The application field they explicitly mention in the introduction is the delivery of ultrashort pulses in optical fibers for materials processing. The motivation to have faster ways of solving the NLSE is the opportunity of a faster analysis of NL effects and thus being able to use less costly fused silica fibers.
-> It took me quite a while to understand this argumentation: the idea is not clearly presented and the way of arguing is quite confusing.
-> The mathematical and physical derivations of the calculation methods seem sound to me. However I lack the possibility of counterchecking the metods in detail e.g. by numerical calculations, as I have no access to the produced programming codes.
-> If the main motivation for the development of such an elaborated solution path to the NLSE is UP delivery, why don’t the authors present more calculation results on this application that would give the workers in the field the possibility to compare the method with their own results?
-> The language used by the authors is not always clear. The paper language should be checked by a scientist of English mother tongue.
-> Detail question : what do the authors mean by « heat connectivity » (line 262) ?
-> Do the authors consider to make their program codes publicly accessible ?
Author Response
First of all, I would like to express our deep gratitude for the careful and scrupulous review of our work. The main idea of our research work was to suggest a numerical algorithm for coupled NLSE solving. The experimental data of N.Karasawa’s and S.Nakamura’s research group allowed us to verify our numerical algorithm and ensured us that our numerical results were in good correlation with their experimental results. That is why we tested our algorithm on ultrashort pulse propagation. We assume, that our algorithm can be also applied to other tasks.
We described the task initial conditions for calculations in section 9.3. It allows other researchers to obtain a solution using their own methods with the same input data and to compare their results with ours. A researcher can make calculations according to our algorithm, which is described in detail in the manuscript.
In general, we agree that the comparison of the NLSE solution methods is of great interest, however, it is a very complicated and non-trivial task, which requires serious analysis and new publication. The present work is devoted to the new method presentation, description, and validation, we also wanted to demonstrate its possibilities on model task solutions. In our opinion there are enough calculations to achieve this goal. The calculation results increasing will not enrich the manuscript and will not make it more innovative, it can only increase its volume. Moreover, it can distract readers from the new method description.
You are right that in line 262 typing mistake occurred, we corrected it for «heat conductivity».
Unfortunately, we are not yet ready to open our source codes, because the registration of copyrights is in progress. After that we will consider the ability of source code opening. Now we are ready to cooperate with interested in NLSE scientists for our method verification in other applications.
Reviewer 2 Report
The draft has studied a numerical method to simulate the pulse propagation in optical fiber using coupled nonlinear Schrodinger equations, which is particular important for a birefringent fiber. The model has considered up to the third order dispersion and Raman scattering. The equations are written in a way that linear terms are written in an implicit scheme and nonlinear terms are written in an explicit finite-difference scheme. It is numerical solved based on finite-difference methods numerical integration algorithm. I think the draft is written well and presented clearly. It should be interesting for broad readership of Fiber. I would like to recommend the publication. But I still have some questions.
1 The authors have said that the method is absolute stable. How about its convergence? What is the smallest step for finite difference?
2 What is the wavelength of the pulse mentioned on page 2?
3 on page 11, any reference for classic heat diffusion equation solutions?
Author Response
First of all, I would like to express my deep gratitude for the careful and scrupulous review of our work. Thank you for giving a positive assessment of our work, it is very important for us.
You are right, that the stability and convergence tasks are very important for any numerical methods and algorithms, especially for non-linear tasks, where there is no possibility to obtain the stability task solution. Nevertheless, we have some materials for choosing integration step. We plan that it will be the theme of our next publication, because it is a large task. Besides, it is well known from the literature, that all implicit symmetrical finite difference schemes are absolutely stable. The additional proof of our method stability is the fact that the solution in section 9.3 was obtained after 720000 integration steps with 20000 time division points. Moreover, our solution is in good correlation with N. Karasawa’s and S.Nakamura’s group experimental results.
The pulse wavelength (Page 2) corresponds to frequency ω0=2.3612×10-15 s–1 and equals the wavelength 798 nm.
The reference [47] to the "Crank, J., Nicolson, P. A practical method for numerical evaluation of solutions of partial differential equations of the heat-conduction type. Mathematical Proceedings of the Cambridge Philosophical Society, 1947, 43(1), 50-67. doi:10.1017/S0305004100023197" for classic heat diffusion equation solutions was given.
Thank you again for your positive review.
Reviewer 3 Report
The nonlinear wave propagation is an important field of physics where numerical simulation provides the powerful means for the investigation. In this paper the new method for solving of the coupled nonlinear Schr\"{o}dinger equations describing two orthogonally polarized modes propagating in a birefringent fiber with taking into account the third-order chromatic dispersion and Raman scattering is presented. The method has a number of advantages over familiar split-step method. That is principal result of this manuscript.
The paper contains mathematical results that may be representing interest for experts in area of the nonlinear optics and the nonlinear wave theory. I believe the manuscript is suitable for publication.
Author Response
First of all, I would like to express my deep gratitude for the careful and scrupulous review of our work. We want to express our gratitude for this. The fact that you positively and highly appreciated the result of our work is very important for us.